# Effects of Dietary or Supplementary Micronutrients on Sex Hormones and IGF-1 in Middle and Older Age: A Systematic Review and Meta-Analysis

**DOI:** 10.3390/nu12051457

**Published:** 2020-05-18

**Authors:** Ryan Janjuha, Diane Bunn, Richard Hayhoe, Lee Hooper, Asmaa Abdelhamid, Shaan Mahmood, Joseph Hayden-Case, Will Appleyard, Sophie Morris, Ailsa Welch

**Affiliations:** 1Norwich Medical School, University of East Anglia, Norwich Research Park, Norwich, Norfolk NR4 7TJ, UK; ryan@janjuha.co.uk (R.J.); r.hayhoe@uea.ac.uk (R.H.); l.hooper@uea.ac.uk (L.H.); asmaa.abdelhamid@uea.ac.uk (A.A.); shaan.mahmood@nuh.nhs.uk (S.M.); joehaydencase@outlook.com (J.H.-C.); will.appleyard@outlook.com (W.A.); sophierose64@gmail.com (S.M.); 2School of Health Sciences, University of East Anglia, Norwich Research Park, Norwich, Norfolk NR4 7TJ, UK; d.bunn@uea.ac.uk

**Keywords:** micronutrients, sarcopenia, sex hormones, insulin-like growth factor 1, meta-analysis, randomized controlled trials

## Abstract

Observational research suggests that micronutrients may be protective for sarcopenia, a key health issue during ageing, potentially via effects on hormone synthesis and metabolism. We aimed to carry out a systematic review of RCTs investigating effects of increasing dietary or supplemental micronutrient intake on sex hormones and IGF-1 in individuals aged 45 years or older. We searched MEDLINE, EMBASE and Cochrane databases for RCTs reporting the effects of different micronutrients (vitamins A, C, D, or E; carotenoids; iron; copper; zinc; magnesium; selenium; and potassium) on sex hormones or IGF-1. Of the 26 RCTs identified, nine examined effects of vitamin D, nine of multi-nutrients, four of carotenoids, two of selenium, one of zinc, and one of vitamin E. For IGF-1 increasing vitamin D (MD: −0.53 nmol/L, 95% CI: −1.58, 0.52), multi-nutrients (MD: 0.60 nmol/L, 95% CI −1.12 to 2.33) and carotenoids (MD −1.32 nmol/L; 95% CI −2.76 to 0.11) had no significant effect on circulating concentrations. No significant effects on sex hormones of other micronutrients were found, but data were very limited. All trials had significant methodological limitations making effects of micronutrient supplementation on sex hormones unclear. Further high quality RCTs with physiological doses of micronutrients in people with low baseline intakes or circulating concentrations, using robust methodology, are required to assess effects of supplementation adequately.

## 1. Introduction

Sarcopenia is a major problem, involving loss of skeletal muscle mass and function with age, a process beginning at approximately 40 years in both men and women [1,2,3]. One mechanism for its onset, is the age-related decline in the endocrine system, including the secretion of sex hormones and insulin-like growth hormone-1 (IGF-1) [4]. Recent evidence suggests that certain micronutrients may be protective for sarcopenia, and also important for hormone synthesis and metabolism, particularly during the decrease in endogenous secretion that occurs during aging [4]. This decrease in hormone secretion is also associated with increases in risks of falls, osteoporosis, fractures, cardiovascular disease and all-cause mortality [5,6,7,8,9,10,11,12,13,14,15,16].

The endocrine system decline with age [4] includes a decrease in testosterone concentrations of 0.5%–1% per year in men, and of oestrogen in women, that begins around 30 years of age [17,18]. The decline in testosterone concentrations in men is associated with loss of muscle mass and strength [19,20], and furthermore testosterone/dihydrotestosterone (DHT) supplementation can increase muscle strength [21]. Similarly, oestrogen concentrations, which decline more rapidly during the menopause in women [22], are closely linked to muscle strength [23]. Evidence from randomised controlled trials (RCTs) suggests oestrogen replacement therapy reduces the decline in strength of post-menopausal women [22] via a reduction in ‘FOXO3’ activation and ‘MuRF1’ protein expression [24].

The concentration of other endocrine hormones, (Dehydroepiandrosterone, DHEAS; Sex-Hormone Binding Globulin, SHBG; and Insulin-like Growth Factor-1, IGF-1) are also associated with skeletal muscle [25,26] and may be involved in the aetiology of sarcopenia since circulating concentrations change with age. DHEAS [27], which is converted into the active forms of testosterone and oestrogen, and stimulates production of IGF-1 [18], declines with age and relates to loss of muscle mass and strength [17]. SHBG transports testosterone, oestrogen, and other steroids in the blood, and increases with age thus reducing free testosterone and oestrogen [28,29,30]. The age-related decline in IGF-1 [31] is relevant due to its roles in promoting myoblast proliferation and differentiation, as well as formation of muscle fibres during normal growth, and in response to injury [32]. Alongside improving muscle hypertrophy and strength, IGF-1 also suppresses muscle inflammation and fibrosis, and is associated with skeletal muscle mass and strength [32,33,34,35,36]. Therefore, increasing concentrations of circulating IGF-1 and sex hormones may be potentially beneficial for preventing sarcopenia as well as certain non-communicable diseases and conditions of aging.

Micronutrients are potentially important for sex hormone synthesis and metabolism, particularly during the age-related decline in the endocrine system. Previous research from in-vitro, in-vivo or observational studies found that certain micronutrients including vitamin D [37,38,39,40,41,42,43,44,45,46,47], vitamin E [48,49,50], vitamin A [51], lycopene [52], iron [53], magnesium [54,55,56], selenium [57,58,59,60] or zinc [58,60,61,62,63,64,65,66,67] were associated with either androgen metabolism, testosterone concentrations or SHBG. Associations have also been found between oestrogen and vitamins C, D, E, A, and carotenoids, including lycopene [48,68,69,70,71]. IGF-1 has also been associated with lycopene [72,73,74,75], magnesium [55], selenium [57] or zinc [61,62,76,77], iron/ferritin [78] and copper [79]. These studies indicate the relevance of micronutrients to the endocrine system, although many were in individuals in young adulthood and their effects in older age have been less studied to date.

The mechanisms for the role of micronutrients in synthesis of sex hormones and IGF-1 include the involvement in steroidogenesis, via the involvement of prostaglandins, on the precursors of sex hormones, for vitamins D, E, the carotenoids, zinc and selenium, as well the effects of vitamin C [38,39,40,41,42,43,44,45,46,47,48,49,50,52,60,64,65,68,69,71,79,80,81,82,83,84,85,86], and effects on transporter proteins. Zinc is also an inhibitor of two enzymes, aromatase and 5α-reductase, that are involved in testosterone metabolism [60].

Intake of micronutrients, micronutrient deficiency, as well as protein intake, may be also important in determining the onset of sarcopenia [38,76,77,87,88,89,90,91,92,93,94,95,96]. Recent observational and animal studies found that vitamins C, D, E, and carotenoids and the minerals magnesium, selenium, iron and zinc are relevant to muscle mass and physical performance [76,77,88,89,92,97,98,99]. The mechanisms for the action of these nutrients include involvement in collagen and carnitine synthesis, for vitamin C, activities on skeletal muscle cell differentiation and proliferation, for vitamin D [38] and synthesis of protein and mitochondrial function, for magnesium [38,76,77,87,88,89,90,91,92,93,94,95,96,97].

Further mechanisms for changes in hormones and the musculoskeletal system that occur during aging are the associated increases in low grade circulating inflammatory cytokines and of ROS (Reactive Oxygen Species) [93]. A number of micronutrients act as endogenous antioxidants with the capacity to reduce ROS and circulating inflammatory cytokines. These micronutrients include vitamins A, C, E, the carotenoids [100,101,102,103,104,105], zinc [60,106], magnesium [55] and selenium [57,60]. Therefore, improving intakes or rectifying micronutrient deficiency could potentially affect both the onset of sarcopenia as well as sex hormone and IGF-1 metabolism, via a number of mechanisms, during aging. Improvements in micronutrient intake could be achieved through increased consumption of dietary whole foods, e.g., oranges (rich in vitamin C); or via supplementation of vitamins and minerals, e.g., single component or multivitamin tablets.

We are unaware of any previous systematic reviews that have investigated the importance of micronutrient intakes on sex hormones and IGF-1 in middle and older aged people at risk of sarcopenia. Therefore, given the potential role for micronutrients to influence secretion of these hormones during aging, and the importance of these sex hormones to the aetiology of sarcopenia, we conducted a systematic review (SR) to investigate the effects of dietary or supplemental intake of specified micronutrients and changes in concentration of sex hormones and IGF-1. We included adults aged 45 years or older, since this is the age at which recognisable declines in muscle mass and function, sex hormones and IGF-1 start to occur [107].

## 2. Materials and Methods

The systematic review was conducted in accordance with the Cochrane collaboration guidelines and reported using the PRISMA 2009 checklist [108,109]. The protocol was registered with the International prospective register of systematic reviews (PROSPERO), registration ID: CRD42018098657 [110].

### 2.1. Search Methods

Cochrane Central Register of Controlled Trials (CENTRAL), MEDLINE, and EMBASE were searched to 2nd April 2019 using the ‘Population, intervention, comparators, outcomes, study design’ (PICOS) Framework (see Table 1) without date restrictions. The search strategy can be viewed in Appendix A, but in brief, a MEDLINE search was developed and adapted for EMBASE and Cochrane, and search limiters were used for RCTs as per the ‘Scottish Collegiate Network’ [111].

### 2.2. Eligibility Criteria

We included randomised controlled trials (RCTs) that assessed the effects of additional micronutrients in adults aged at least 45 years on primary outcomes. The primary outcomes were changes or differences in sex hormone concentrations, including: androgens (androstenediol, androstenedione, dihydrotestosterone and testosterone), oestrogens (E2, estradiol, estriol, and estrone), DHEAS, SHBG, and IGF-1 (see Appendix A). Relevant micronutrients were those with known or potential relevance to sex hormone or IGF-1 metabolism and physiology, as well as sarcopenia, and included any one, or combination, of vitamin A [48,69]; vitamin C [76]; vitamin D [38,39,40,41,42,43,44,45,46,47]; vitamin E [48,50]; carotenoids [69]; or the minerals zinc [64,65], magnesium [54], selenium, potassium [76,77], iron/ferritin [78] and copper [79].

Where studies included groups of individuals with varying age ranges, they were included if the mean age was greater than 45 years, or more than 75% of individuals were older than 45 years (in both treatment arms). We included studies that used any micronutrient or hormone extraction method, including biomarkers from blood, plasma, red blood cells, body fat, urine, hair, and nails. We excluded studies where the age of the population was unclear, or where participants stopped, started, or changed, hormonal medication, during a study. Where RCTs examined micronutrients in conjunction with another intervention, e.g., exercise, the study was included only if the comparator group received the same non-dietary intervention. Studies that included participants on active dialysis, or with kidney or liver disease [112], were excluded as these are known to affect endogenous sex hormones and IGF-1 [113,114]. In-vitro studies and studies that used foods as interventions without a reported dose of an eligible nutrient were also excluded. We accepted trials of multivitamins or multi-nutrient studies that included further compounds, other than the micronutrients previously listed. This is because some studies may have used combined vitamins for an intervention and provided information. Studies that fell into this category were reported separately, and were defined as two or more different multi-nutrients in the intervention group compared to placebo. We excluded non-English language papers that we were unable to translate within the research team.

### 2.3. Study Selection

Study selection was conducted in a two-phase process. Screening of titles and abstracts against inclusion/exclusion criteria (Appendix A) was carried out independently by two reviewers (RJ and one of DB, AA, RH, AW, SM, JC, WA). Potential titles and abstracts identified by any reviewer were collected in full text and subsequently assessed against the inclusion/exclusion criteria by at least 2 reviewers. Any disagreements were discussed, a third reviewer was not needed to clarify consensus on eligibility.

### 2.4. Data Extraction

We created and tested a data extraction form for this review (Appendix A). Data extracted included: publication details, aims, objectives, country, setting, design, dates, funding, recruitment method, ethical review, participant demographics, intervention descriptions (including: micronutrient type and extraction methods) and outcomes (method of extraction and hormone type). Data extraction and risk of bias assessment was completed independently in duplicate by RJ and another review team member. We were unable to contact authors on any queries regarding data due to time constraints.

### 2.5. Risk of Bias (Quality Assessment)

Risk of bias assessment was based on the Cochrane Risk of Bias tool (https://handbook-5-1.cochrane.org/chapter_8/table_8_5_a_the_cochrane_collaborations_tool_for_assessing.htm) (Appendix A) [115]. Alongside the typical seven standard domains, we included three further items: ‘hormonal treatment’ bias, where participants may be taking medication(s) that influence sex hormones; ‘sponsorship’ bias, where funding by companies may have influenced the outcome of results; and ‘outcome measurement’ bias, which concerns the differences in accuracy of extraction methods. The Journal of Clinical Endocrinology and Metabolism [116] recommends measurement of sex hormones to be conducted using mass spectrometry, as this conveys the highest degree of accuracy, and lowest bias. Studies that used other (less reliable) methods of hormone extraction, e.g., direct immunoassay [117] or electro chemiluminescent assay [118], were assessed to be at high risk of outcome measurement bias.

### 2.6. Data Synthesis and Statistical Analysis

Meta-analyses were performed only where at least two trials could be combined. We used a random effects model in ‘Review Manager (RevMan) [Computer program] [119]. We produced the forest plots using ‘end data’ for intervention and placebo groups. Outcomes were ‘continuous’ and data for IGF-1, testosterone, and SHBG, reported in non-standard units were converted using an online tool (http://unitslab.com/node/230). Where meta-analysis was not possible or data could not be converted or utilised, results were narratively reported. Some studies reported data as ‘median’ values so could not be included in a meta-analysis, but have been included in some forest plots to help illustrate overall effects. Sensitivity analysis, using fixed-effects meta-analysis, was carried out where at least two trials were combined. Comparison between random and fixed effects meta-analysis allowed small study bias to be assessed [120]. We also intended to use funnel plots to assess small study (publication) bias; but as no meta-analysis included at least 10 studies, this was not useful.

## 3. Results

A total of 7623 titles and abstracts were identified from the three separate databases. After the removal of duplicates, 5444 papers remained, of which 5043 were excluded based on title and abstract screening. The remaining 400 studies were assessed in full text, leaving a total of 26 eligible studies. The majority of excluded studies (*n* = 374) were excluded due to the population age or study design. A summary overview of the selection process is provided in Figure 1.

The 26 eligible RCTs examined a range of micronutrients: vitamin D (9, 35%), multi-nutrients (9, 35%), carotenoids (4, 15%), selenium (2, 8%), vitamin E (1, 4%) and zinc (1, 4%). Briefly, a total of 2443 participants were examined. Interventions ranged from 4 weeks to 48 months and participants were mostly males (~64%). Some studies (9, 35%) included individuals who either had a histological diagnosis of prostate cancer or colon cancer, evidence of increasing prostate specific antigen (PSA), or a family history of cancer. Other studies (8, 31%) examined individuals with metabolic syndrome (including obesity) and/or cardiovascular disease. Different races/ethnic groups were also studied, including: Asian, Black, Latino and White. Details of the study characteristics can be found in Appendix A. An overview of the risk of bias for RCTs is shown in Figure 2. We found no trials assessing effects of vitamins A or C, potassium, iron or copper on our outcomes.

### 3.1. Risk of Bias of Included RCTs

Methods of minimising selection bias were poorly reported, with 54% (14/26) and 77% (20/26) of RCT studies being unclear in methodology of ‘randomisation’ and ‘allocation concealment’, respectively (Figure 2). Many studies (58%, or 15/26) minimised performance bias by blinding participants and study personnel, and a smaller proportion (42%, or 11/26) successfully reported blinding of outcomes. Incomplete outcome data was minimised, as was the influence of hormonal treatments on sex hormones (mainly through comprehensive exclusion criteria). Only a small proportion of studies (~12%) advised participants to change their dietary habits in addition to any intervention or placebo. The majority of studies (88%, or 23/26) reported serum blood concentrations, with 12% (3/26) estimating micronutrient intake from dietary assessment questionnaires. Although extraction using serum analysis for micronutrients may pick up coagulants and other trace elements, there appears to be a non-significant variation between plasma and serum values. It is unclear whether the coagulants or trace elements would influence supplemented or non-supplemented cohorts differently. It is worth noting, Olmedilla-Alonso et al. [121] found retinol, gamma- and alpha-tocopherol serum values were positively biased (mean difference of less than: 0.05, 0.01 and 0.7 µmol/L, respectively) when compared to plasma values [121]. However, this is unlikely to influence our results as we only identified one trial with vitamin E within our systematic review. Only 8% (2/26) of studies measured sex hormones using the gold standard recommendation of mass spectrometry [122].

### 3.2. Vitamin D

Nine studies assessed effects of vitamin D on relevant outcomes [116,123,124,125,126,127,128,129,130] but no studies assessed effects on androstenediol, androstenedione, dihydrotestosterone, estriol, or DHEAS. Vitamin D doses varied from 100 IU [130], through 1000 IU [125], 4000 IU [129], 20,000 IU [128] up to 40,000 IU [123], and one was unclear [126]. Baseline vitamin D status was low in some trials [128,129], normal in some [127] and unknown in others [123,124]. Study duration ranged from 6 weeks [130] through 1 year [116,123,125], up to 36 months [129].

#### 3.2.1. Effects of Vitamin D on IGF-1

Four studies [123,124,125,131] assessed the effects of vitamin D supplementation on IGF-1 over 4 weeks to 12 months. We presented the Kamycheva trial [123] as two groups, severely obese (study participants with >35 kg/m^2^) and non-severely obese (other participants), as results were presented this way in the paper. Meta-analysis demonstrated no significant effects of the intervention (mean difference: −0.53 nmol/L, 95% CI: −1.58, 0.52, 3 RCTs, I2 0%, Figure 3). One trial [124] could not be included in the meta-analysis because it was not possible to convert the units of IGF-1 used (μg/10E06 platelets) to nmol/L. This was a 4-week RCT that confirmed a statistically non-significant mean difference of 0.007 μg/10E06 platelets, *p* = 0.413 between intervention and placebo post intervention. The four included trials randomised 447 participants (mean age: 55.2, 59% males, including dropouts) from Norway [123], USA [124,125] and Austria [131]. Studies used a variety of vitamin D dosages: 400 IU [124] 1000 IU [125], 2800 IU [131] and 40,000 IU [123].

Major sources of bias within these studies included randomisation procedures (sequence generation and allocation concealment) and blinding (Figure 3). Only one trial was at low risk of attrition bias, and no studies used ‘mass spectrometry’ to measure sex hormone concentrations, so all were at high risk of outcome assessment bias. 

The lack of effect of increasing vitamin D on IGF-1 was confirmed in the set of trials which supplemented with vitamin D and other compounds (two or more micronutrients) (Figure 3). Combining all the trials increasing vitamin D (individually or as part of a broader intervention) suggests little or no effect on IGF-1 (MD: −0.27 nmol/L, 95% CI −1.20 to 0.67, I2 0%). This did not differ in sensitivity analysis using fixed-effects meta-analysis (MD: −0.27 nmol/L, 95% CI −1.20 to 0.67, I2 0%). The difference in effect size between fixed- and random-effects meta-analysis suggests that there may be some small study bias present.

The effect of differing baseline vitamin D status, doses and study duration were assessed in sub-grouping. There were no differences between subgroups in any analysis (*p* ≥ 0.85 for all subgroupings, not shown).

#### 3.2.2. Effects of Vitamin D on Testosterone

Five trials reported effects of vitamin D on testosterone. They included 754 participants (54% male) from China [126], The Netherlands [127], Austria [128], Germany [129] and USA [116]. All included women were post-menopausal. Only two studies could be combined in meta-analysis, suggesting no effect of vitamin D on free testosterone (MD 0.00, 95% CI −0.00 to 0.00, I2 0%, Figure 4). The effect did not differ in sensitivity analysis using fixed-effects meta-analysis, suggesting a lack of small study bias, although with only two trials this is difficult to assess. The other trials (shown in Figure 4 though not combined in meta-analysis) reported data as medians and interquartile ranges [127,128]. One study did not specify which type of testosterone was measured and did not provide enough data to be included [126].

The two trials that could be combined appeared to have adequate randomisation and blinding, though only one had adequate allocation concealment. Neither was at a low risk of outcome measurement bias.

#### 3.2.3. Effects of Vitamin D on Oestradiol

Two trials assessed effects of vitamin D on oestradiol, but could not be combined in meta-analysis. Individually, neither found a statistically significant effect of supplementation [116,126].

#### 3.2.4. Effects of Vitamin D on SHBG

Three studies reported effects of vitamin D on SHBG, randomising 445 participants (51% male) [116,128,129]. Meta-analysis of two RCTs suggested a small though non-statistically significant increase in SHBG with vitamin D (MD 4.18 nmol/L, 95% CI −1.28 to 9.64, I2 0%, Figure 5), but data from the third trial contradicted this finding. Two of the three trials were at low risk of selection bias, and all were well blinded, and one used a low risk method of outcome assessment (Figure 5).

### 3.3. Multi-Nutrients

Nine studies assessed effects of multi-nutrients (defined as two or more different micronutrients in the intervention group) on relevant outcomes, but no studies assessed effects on androstenediol, androstenedione, E2, estradiol, estriol, estrone, or DHEAS. Those studies that also included vitamin D within the multi-nutrient interventions are also covered in Section 3.2.

#### 3.3.1. Effects of Multi-Nutrient Supplements on IGF-1

Seven of the nine studies identified as multi-nutrient interventions [130,132,133,134,135,136,137] reported on IGF-1, suggesting little or no effect (MD: 0.60 nmol/L, 95% CI −1.12 to 2.33, I2 0%, 519 participants, Figure 6). Effects did not differ in fixed-effects meta-analysis (MD: 0.60 nmol/L, 95% CI −1.12 to 2.33), suggesting that small study bias is not an issue here.

The Jensen et al. study [137] analysed data using a ‘per-protocol’ method which introduced potential bias [138], since the 20% of study participants withdrew. All seven studies measured micronutrient concentrations from blood samples, but none used mass-spectrometry to measure sex hormone concentrations, and many were unclear on selection bias and blinding.

#### 3.3.2. Effects of Multi-Nutrients on Testosterone

Two trials carried out in the Netherlands reported effects of multi-nutrient interventions on testosterone in men with rising levels of prostate-specific antigen (117 males, mean age: 72), and reported opposing findings (Figure 6). We were unable to meta-analyse these findings as Kranse did not provide any measure of variance. However, Hoenjet, 2005 [139] suggested no effect on testosterone concentrations (*p* = 0.28), while Kranse, 2005 [140] (cross-over trial) suggested significant reductions in testosterone but reported different numbers in different places in their paper, so the effect size was unclear (*p* = 0.02). Both trials were at unclear risk of selection bias and low risk of blinding problems.

#### 3.3.3. Effects of Multi-Nutrients on Dihydrotestosterone and SHBG

The same two trials (Kranse, 2005; and Hoenjet, 2005) also assessed the effect of multi-nutrient supplementation on Dihydroteststerone and SHBG (Figure 6), but again Kranse provided no measure of variance and two different effect sizes, so could not be pooled. Multi-nutrient supplementation in Kranse, 2005, reportedly significantly decreased Dihydrotestosterone, but the effect size was unclear (*p* = 0.005), whereas, in Hoenjet 2005, non-significant findings were reported (MD: 0.1 nmol/L, 95% CI −0.1 to 0.2, *p* = 0.72). After supplementation with multi-nutrients, both studies reported non-significant decreases in SHBG.

### 3.4. Carotenoids

Four trials assessed the effects of carotenoids on relevant outcomes, but none assessed the effects on androgens (androstenediol, androstenedione, dihydrotestosterone or testosterone), oestrogens (E2, estradiol, estriol, or estrone), DHEAS or SHBG.

#### Effects of Carotenoids on IGF-1

Four studies [52,80,81,82] examined the effects of lycopene, all using ‘Lyco-O-Mato’ (containing ~15 mg lycopene, plus 1.5 mg phytoene, 1.4 mg phytofluene, 0.4 mg beta-carotene, and 5 mg alpha tocopherol). Meta-analysis of 278 randomised participants (mean age: 63.0, 75% male) showed a non-significant decrease in IGF-1 as a result of the added carotenoids (MD −1.32 nmol/L; 95% CI −2.76 to 0.11, I2 0%, Figure 6). None of the trials were at low risk of selection bias, but two were at low risk from issues around blinding (performance and detection bias), and none used mass-spectrometry to measure hormone concentrations (Figure 7).

Trials of carotenoids as part of multi-nutrient supplementation (Figure 7), confirmed a small non-significant decrease in IGF-1 (MD: −0.39, 95% CI −2.90 to 0.11, I2 0%). These trials also demonstrated significant sources of bias (see earlier).

Overall effects of carotenoids (in either individual or multi-nutrient studies) suggested no important effect of carotenoids on IGF-1 (MD: −1.09, 95% CI −2.34 to 0.16, I2 0%), which did not differ in sensitivity analysis using fixed effects meta-analysis (MD: −1.09, 95% CI −2.34 to 0.16). This suggested minimal small study bias.

### 3.5. Selenium

Two studies assessed effects of selenium on testosterone, but no studies assessed effects on oestrogens (E2, estradiol, estriol, or estrone), DHEAS, SHBG, IGF-1 or androgens other than testosterone.

#### Effects of Selenium on Testosterone

Two studies in the Czech Republic examined the effects of 240 μg of selenium (as selenomethionine) on testosterone. Both intervention and placebo also received 570 mg of silymarin, an extract of milk thistle [141,142]. Both suggested no significant effects on testosterone.

### 3.6. Vitamin E

One study assessed the effects of Vitamin E on DHEAS, but no studies assessed the effects on androgens (androstenediol, androstenedione, dihydrotestosterone and testosterone), oestrogens (E2, estradiol, estriol, and estrone), SHBG, or IGF-1.

#### Effects of Vitamin E on DHEAS

Amsterdam 2005 [143] found that 200 mg vitamin E (as dl-alpha-tocopheryl acetate) over 15 months lead to a significant decrease in DHEAS in the supplemented group (*p* < 0.02) but not the placebo group (*p* > 0.05). The authors concluded there was no overall significant benefit to vitamin E supplementation.

### 3.7. Zinc

One study assessed effects of zinc on IGF-1, but no studies assessed effects on androgens (androstenediol, androstenedione, dihydrotestosterone and testosterone), oestrogens (E2, estradiol, estriol, and estrone), DHEA or SHBG.

#### Effects of Zinc on IGF-1

A Swiss trial by Rodondi, 2009 [61] (*n* = 69, mean age 85, 86% female) reported that supplementation of 30 mg/day of zinc (alongside 15 g whey protein + 5 g amino acids) increased serum IGF-1 over a week compared to protein alone (+48.2% vs. +22.4%, respectively; *p* < 0.027), but there was no statistically significant difference between groups at 4 weeks (+29.2% vs. +45.8%; *p* > 0.05).

## 4. Discussion

We found 26 trials assessing effects of micronutrient supplementation, but no trials assessing effects of vitamins A or C, potassium, iron, or copper, on our outcomes. Data from nine trials suggested that supplementation with vitamin D had little or no effect on IGF-1, with or without other micronutrient compounds. Vitamin D also failed to significantly alter testosterone or oestradiol, and the effects on SHBG and other outcomes were unclear. The multinutrient trials did not suggest statistically significant increases in IGF-1, and the effects on testosterone, dihydrotestosterone and SHBG were unclear. Data were very limited for effects of other micronutrients. Four trials suggested that carotenoids slightly reduce IGF-1 and this was reinforced with the inclusion of other micronutrients (though none of the relationships were statistically significant). Selenium appears to have little effect on testosterone (2 trials), vitamin E had no effect on DHEAS, and zinc had little or no effect on IGF-1 (a single trial each).

Despite our systematic search including a large range of relevant micronutrients and hormones, we only identified studies that investigated effects of vitamin D, multi-nutrients, the carotenoids, selenium, vitamin E, and zinc, on sex hormones and IGF-1. To the best of our knowledge, this is the first systematic review examining the relationship between this range of micronutrients, sex hormones and IGF-1 in people of middle and older age. We conducted the review using established Cochrane methodology [115].

Despite the biochemical, physiological and mechanistic roles of micronutrients for hormone synthesis in older age our review found a paucity of trials and little direct evidence of significant effects of micronutrient supplementation [38,39,40,41,42,43,44,45,46,47,48,49,50,52,60,64,65,68,69,71,79,80,81,82,83,84,85,86]. Since the age-related decline in sex-hormones and IGF-1 not only increases the risk of sarcopenia, but also a number of conditions of aging, including falls, osteoporosis, fractures, cardiovascular disease and all-cause mortality, this is unfortunate [5,6,7,8,9,10,11,12,13,14,15,16].

### Limitations of the Available Data

Whilst we identified 26 RCTs of adults aged at least 45 years that met our eligibility criteria, when grouped by micronutrient and sex-hormone, the number of studies in each category was small (between one and nine studies per nutrient), and many had methodological limitations. A number of the studies had small sample sizes or lack of control for dietary or lifestyle determinants in the intervention and control groups [52,80,82] and one study [137,138] also analysed data using a ‘per-protocol’ method which may have introduced bias elements of bias [138].

For multi-nutrient interventions the composition of the nutrients varied substantially [130,132,133,134,135,136] with some containing more than 20 different micronutrients [132,137], making it difficult to attribute benefit to any specific micronutrient. A number of studies also included additional protein making it difficult to isolate any specific effects of micronutrients from those of protein [61,132,137,144,145]. The baseline nutritional status of participants was not taken into account in a number of studies despite baseline status or deficiency being likely to determine the response to interventions. Some studies also included dietary advice to increase sources of calcium, which may have affected the results of the intervention.

Whilst we found no significant effect of the supplementation of micronutrients on circulating sex hormones and IGF-1, the scale of effects for the few studies that included IGF-1 ranged between mean differences for vitamin D of −0.53 nmol/L (95% CI: −1.58, 0.52), for multi-nutrients of 0.60 nmol/L (95% CI −1.12 to 2.33) and carotenoids of −1.32 nmol/L (95% CI −2.76 to 0.11). This compares with the difference for IGF-1 between age groups 50–54 years to 70–74 years of −3.4 nmol/L [146,147]. Although the effect sizes found with micronutrients and IGF-1 in our analysis were non-significant, and smaller than with age, these differences may have potential importance if found to be significant in future well-designed trials.

## 5. Recommendations for Future Studies

Although our systematic review demonstrated no conclusive effects of the supplementation of micronutrients on sex hormones in middle- and older-aged people, we recommend that larger RCTs are conducted specifically targeting the micronutrients where we found little or no existing research (magnesium, zinc, vitamins A, C and E, iron, copper and potassium). Future RCTs should be of sufficient size and include baseline and follow-up measures of dietary intake (such as with food frequency questionnaires), as well as using blood concentrations of the relevant micronutrients. This would clarify whether micronutrient supplementation is only beneficial to depleted individuals or whether it can provide additional benefit to those with adequate micronutrient status. Direct measurements of micronutrient status have advantages as they are independent of potential reporting bias, are integrated measurements of intake and other physiological and lifestyle influences on status, such as smoking habit, and can be used to determine whether supplementation results in improved micronutrient status [148,149,150]. Furthermore, dosages of micronutrients should be designed to rectify any pre-existing micronutrient deficiency. Additionally, extraction of hormones should be performed using mass spectrometry, and SHBG should be measured to account for changes to free oestrogen and testosterone that may occur during the intervention. Other known lifestyle factors that affect circulating sex hormones and IGF-1, such as smoking habit and BMI should also be recorded [146,147]. An optimal follow-up time has yet to be elucidated but we would recommend a minimum of 6 months, and that endocrine and nutritional measurements be taken at 3 month intervals until the study is complete.

## 6. Conclusions

Effects of micronutrient supplementation on sex hormones and IGF-1 are unclear. Further high quality RCTs with physiological doses of micronutrients in people with low baseline intakes or circulating concentrations, using robust methodology, are required to assess effects adequately.

## Figures and Tables

**Figure 1 nutrients-12-01457-f001:**
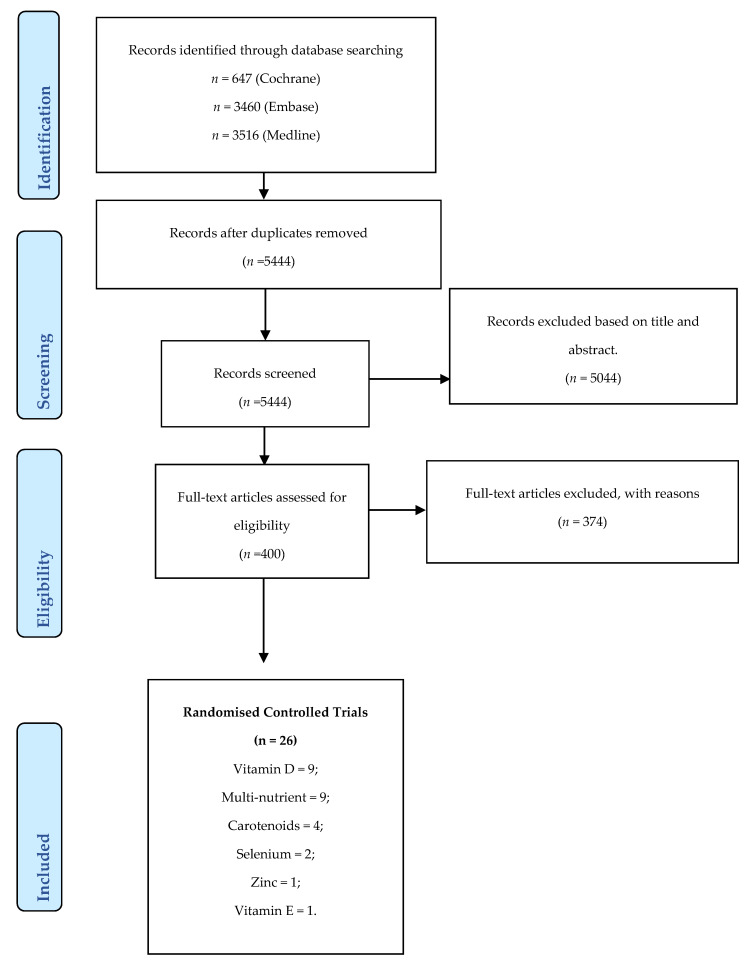
PRISMA Flowchart.

**Figure 2 nutrients-12-01457-f002:**
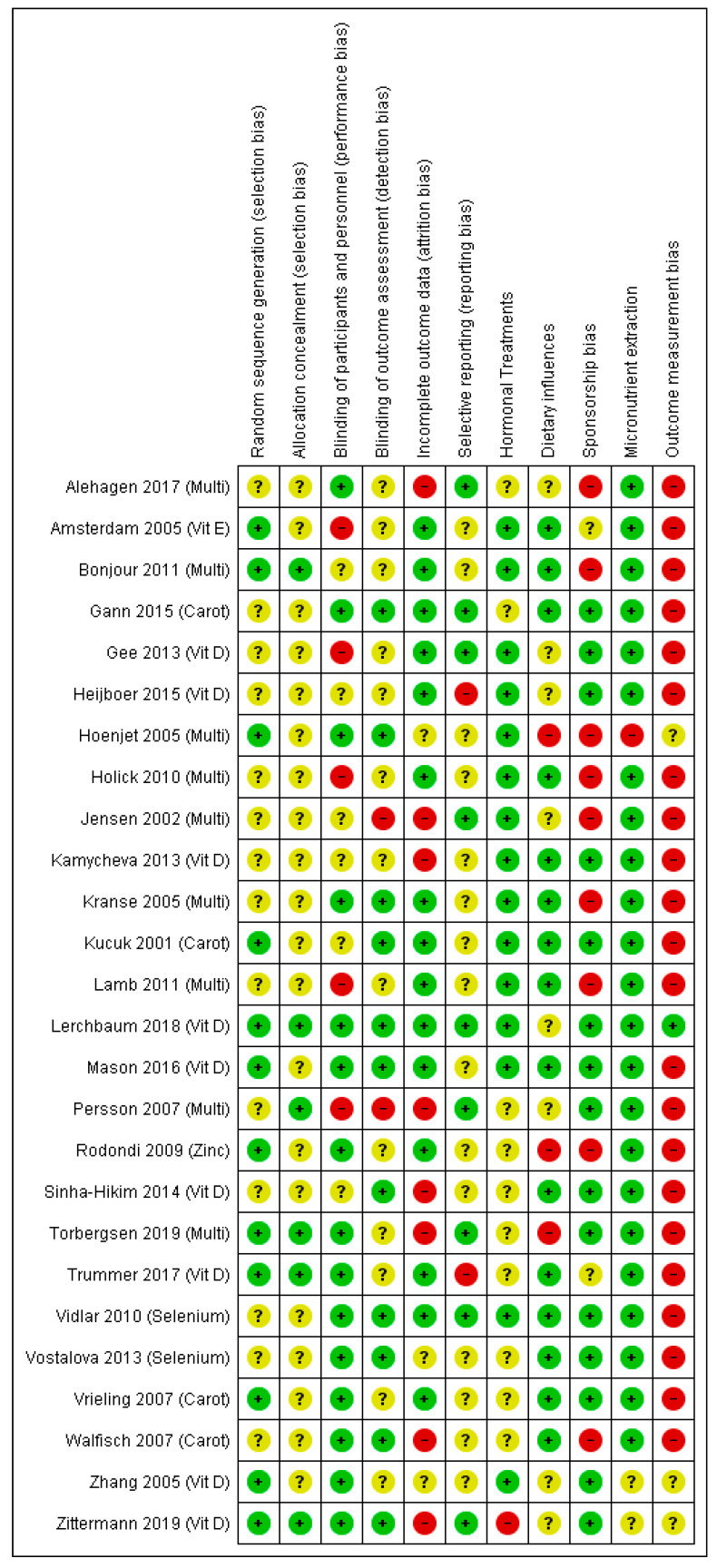
Risk of bias of included RCTs, assessed for each domain and each included trial, based on the Cochrane Risk of Bias tool [120]. +: low risk of bias, ?: unclear risk of bias, -: high risk of bias. Carot, carotenoids; multi, multi-nutrient; vit D, vitamin D; vit E, vitamin E. Please refer to Appendix A to find detailed information on the studies and reference details.

**Figure 3 nutrients-12-01457-f003:**
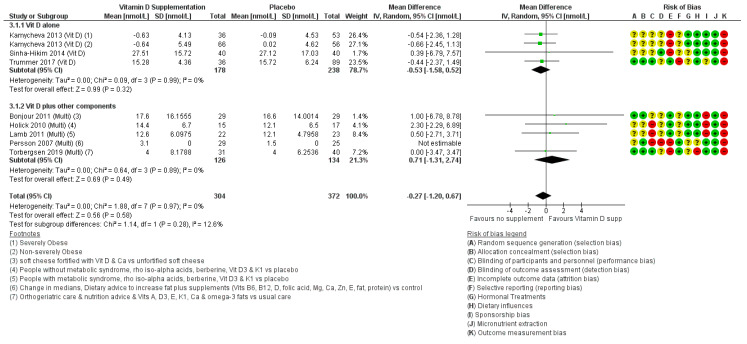
Forest plot assessing effects of increasing vitamin D intake and vitamin D amongst other nutrients, on IGF-1 (nmol/L). Multi, multi-nutrient; vit D, vitamin D. Please refer to Appendix A to find detailed information on the studies and reference details.

**Figure 4 nutrients-12-01457-f004:**
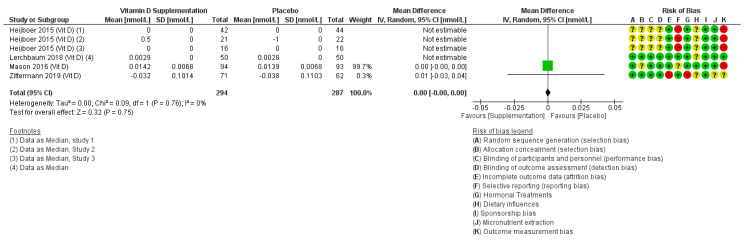
Forest plot assessing effects of increasing vitamin D on free testosterone (nmol/L). Vit D, vitamin D. Please refer to Appendix A to find detailed information on the studies and reference details.

**Figure 5 nutrients-12-01457-f005:**
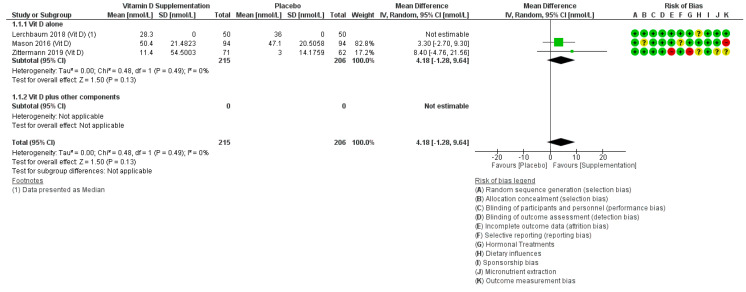
Forest plot showing effects of increasing vitamin D on SHBG (nmol/L). Vit D, vitamin D. Please refer to Appendix A to find detailed information on the studies and reference details.

**Figure 6 nutrients-12-01457-f006:**
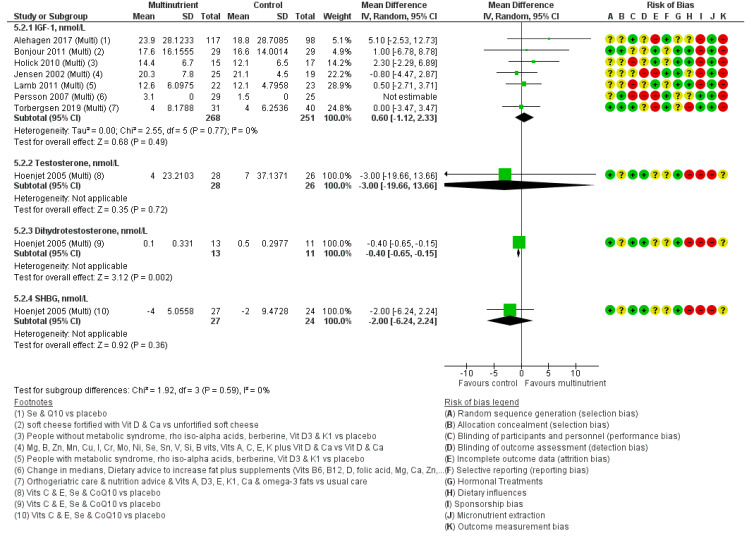
Forest plot assessing the effects of multi-nutrient interventions on sex hormones and IGF-1 (nmol/L). Multi, multi-nutrient. Please refer to Appendix A to find detailed information on the studies and reference details.

**Figure 7 nutrients-12-01457-f007:**
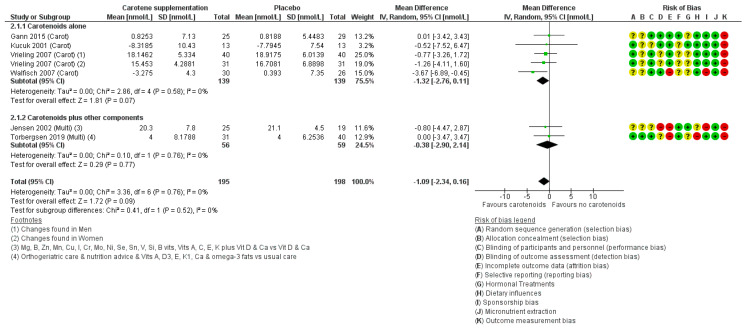
Forest plot assessing effects of increasing carotenoids (lycopene) and carotenoids amongst other nutrients on IGF-1 (nmol/L). Carot, carotenoids; multi, multi-nutrient. Please refer to Appendix A to find detailed information on the studies and reference details.

**Table 1 nutrients-12-01457-t001:** PICO Framework for search strategy. See Appendix A, for further details.

**P**	Humans, adults only, aged >45 years.
**I**	Micronutrients
**C**	-
**O**	Sex hormones and IGF-1
**S**	Randomised controlled trials (RCTs)

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
