# Peer review of "Effects of Dietary or Supplementary Micronutrients on Sex Hormones and IGF-1 in Middle and Older Age: A Systematic Review and Meta-Analysis"

_nutrients, 2020, doi:10.3390/nu12051457_

Round 1

Reviewer 1 Report

This is a meta-analysis of an important topic, with a lot of work done. There are, however, a few changes to be seriously considered.

The mentioning of sarcopenia seems out of place in the introduction. I agree that the effects of dietary supplements on sex hormones and IGF-1 are important to understand, especially for middle and older age people. However, it doesn't flow well when the authors put the emphasis on sarcopenia only in the introduction, while throughout the manuscript, no further discussion was relevant to sarcopenia. I suggest that the authors discuss more broadly re other health outcomes.

1. Line 189: Although the details of the included studies could be found in supplementary table E, it would be good to briefly describe e.g. range of trial durations, population sizes, race/ethnic groups, etc. This is to give the readers a general idea of their similarities and major differences.

2. Line 234: 'skewing true biochemical values for macronutrient status' - how? and how would coagulants and other trace elements influence the contrast between takers and non-takers in the same study? Please specify.

3. Line 245: were the subgroups defined based on individual-level BMI or group-level BMI (e.g. median)? also, I don't see results (for example, p for interaction w/ BMI) that 'help illustrate findings'.

4. Other than the narrative summary and risk-of-bias assessment, was there any analysis done such as the test for publication bias, small study effect, trend by publication year?

Overall, I personally feel that the results and discussion could be more focused on the 'main findings', for example, figure 3, figure 6, and figure 7. Since the authors summarized results for many micronutrients and many hormones, it seems a little all over the place if all synthesis of ≥2 studies were presented. The goal of any meta-analysis is to best summarize available evidence and convey a clear and conclusive message. For this study specifically, given that the risk-of-bias are quite concerning, I suggest that the authors decided on fewer exposures and outcomes that have enough evidence to draw a conclusion.

Reviewer 2 Report

The authors have drafted a well written manuscript clearly describing the study's design, selection and eligibility criteria, filtering parameters, and statistical analyses. Findings on the meta-analyses of the RCT's reported revealed subtle direct effects of micronutrient on sex hormone or IGF-1 levels, which the authors acknowledged arose from limitations regarding methodology including sample size, differences in RCTs micronutrient composition, as well as several other RCTs experimental confounds. The authors recommendations and conclusions are indeed supported by the data reported.

Reviewer 3 Report

The authors of the meta-analysis manuscript evaluate the effect of different micronutrients on the concentration of IFG-1 and of sex hormones in adults. The manuscript is an attractive methodological proposal and could add interesting data to the body of knowledge about micronutrients and aging, although, as the authors indicate, the effects of micronutrients on changes in these molecules are complex to analyze and systematize. The procedures are well described and the results correctly presented. Despite the authors finally conclude that no effects of vitamin D, multi-nutrients, carotenoids, and selenium are observed, this work seems interesting to readers when it comes to supporting the knowledge about supplemental micronutrient intake or increasing dietary micronutrients in the elderly population. The conclusion, clinical trials are needed, is a reminder to researchers to carry out deeper, well-designed and reported studies. 

Reviewer 4 Report

I have read the article entitled “Effects of dietary or supplementary micronutrients on sex hormones and IGF-1 in middle and older age: a systematic review and meta-analysis” by Janjuha et al. The authors present a systematic review of RCTs investigating effects of increasing dietary or supplemental micronutrient intake on sex hormones and IGF-1 in individuals aged 45 years or older. They have searched MEDLINE, EMBASE and Cochrane databases for RCTs reporting the effects of different micronutrients (vitamins A, C, D, or E; carotenoids; iron; copper; zinc; magnesium; selenium; and potassium) on sex hormones or IGF-1. Of the 26 RCTs identified, nine examined effects of vitamin D, nine of multi-nutrients, four of carotenoids, two of selenium, one of zinc, and one of vitamin E. For IGF-1 increasing vitamin D, multi-nutrients and carotenoids had no significant effect on circulating concentrations. No significant effects on sex hormones with other micronutrients were found. The authors conclude that all trials had significant methodological limitations making the effects of micronutrient supplementation on sex hormones and IGF-1 unclear. Further high quality RCTs with physiological doses of micronutrients in people with low baseline intakes, using robust methodology, are required to assess effects of supplementation adequately.

I think that this systematic review and meta-analysis presents an updated and interesting study about the effects of dietary or supplementary micronutrients on sex hormones and IGF-1. I find very adequate the methodological aspects of the article.

Nevertheless, the paper has several limitations, the most important is that I have doubts about the relevance to investigate the effects of dietary or supplemental intake of specified micronutrients and changes in concentration of sex hormones and IGF-1.

The authors did not provide enough evidence to suggest that certain dietary or supplementary micronutrients may be protective for sarcopenia, and also important for hormone synthesis and metabolism. In that sense it is important to clearly differentiate the effect of replacement from supplementation.

Could you explain more clearly the difference between dietary or supplementary micronutrients.

Could you include some evidence that 45 yr. is the age at which declines in muscle mass and function start to occur.

Figure 2 is not explained in the footnote.

Figures 3-7 are not clearly presented and very difficult to read.

Some aspects of the discussion could be improved. I think that you should reinforce both in the introduction and discussion the potential biochemical, physiological and mechanistic roles of micronutrients supplementation for hormone synthesis in older age.

I think that at the present time, as you suggest, we first of all need studies using blood concentrations of the relevant micronutrients, in order to clearly differentiate replacement from supplementation, could you comment more on that aspect.

There are several mistakes along the test, like reference 83.

Round 2

Reviewer 1 Report

The authors have addressed the concerns I had. The manuscript looks nice and clear. Upon final proofread I have no further concerns.

Reviewer 4 Report

Accept in present form